# Investigating Iron Alloy Phase Changes Using High Temperature In Situ SEM Techniques

**DOI:** 10.3390/ma15113921

**Published:** 2022-05-31

**Authors:** Rhiannon Heard, Clive R. Siviour, Kalin Dragnevski

**Affiliations:** 1Solid Mechanics, Engineering Science Department, University of Oxford, Oxford OX1 2JD, UK; clive.siviour@eng.ox.ac.uk (C.R.S.); kalin.dragnevski@eng.ox.ac.uk (K.D.); 2Imaging and Analysis Centre, Natural History Museum, London SW7 5BD, UK

**Keywords:** in situ SEM, EBSD, heat treatment, carbon steel

## Abstract

This research utilises a novel heat stage combined with a Zeiss scanning electron microscope to investigate phase changes in iron alloys at temperatures up to 800 ℃ using SE and EBSD imaging. Carbon steel samples with starting structures of ferrite/pearlite were transformed into austenite using the commercial heat treatment process whilst imaging within the SEM. This process facilitates capturing both grain and phase transformation in real time allowing better insight into the microstructural evolution and overall phase change kinetics of this heat treatment. The technique for imaging uses a combination of localised EBSD high temperature imaging combined with the development of high temperature thermal-etching SE imaging technique. The SE thermal etching technique, as verified by EBSD images, enables tracking of a statistically significant number of grains (>100) and identification of individual phases. As well as being applied to carbon steel as shown here, the technique is part of a larger study on high temperature in situ SEM techniques and could be applied to a variety of alloys to study complex phase transformations.

## 1. Introduction

This paper presents investigations of the phase transformation into the austenitic region during the heat treatment of carbon steel. Investigations using novel in situ high temperature SEM imaging techniques with the use of a new heat stage [1] provide unique insight into the phase evolution of carbon steel from the starting microstructure into the austenitic region. The final microstructure obtained after heat treatment is strongly dependent on the reverse transformation kinetics during austenite formation, which are dictated by the initial microstructure [2]. Hence, this paper presents the studies on the ferrite/pearlite initial starting structures, which generate microstructures with desirable properties, when transformed into austenite during commercial heat treatment processes. 

The transformation of the complex ferrite/pearlite phase structure into austenite is considered a diffusion driven process of nucleation and growth [3], where dilatometric analysis indicates a two-stage process of pearlite dissolution followed by the transformation of the pure ferrite grains into austenite [2]. These findings are further supported by microstructural data captured via interstitial rapid cooling after a short heat treatment, indicating the initial formation position of austenite with respect to pearlite and ferrite grain boundaries [4]. 

Localised in situ microscopy studies of pure ferrite to austenite transformations using optical and laser scanning confocal microscopy, and electron backscatter diffraction (EBSD) imaging have shown the benefit of real-time microstructural observations in supporting dilatometric analysis and providing insight into specific individual grain interface behaviour during the phase change. Nonetheless, these advances are subject to typical instrumental constraints including spatial resolution, imaging speed, and field of view [4]. Due to the complexity of the ferrite/pearlite to austenite phase change, the development and application of a technique that can capture a more statistically significant number of grains in real time would greatly benefit understanding and quantification of the average microstructural movement during this process.

The literature indicates there is a lack of data on specific real-time microstructural evolution during this phase change; in particular, for large statistically significant data sets of multiple grains. This paper uses in situ real-time EBSD and SE data captured during the phase change to further understanding of the microstructural evolution during this transition. 

## 2. Materials and Methods

A high temperature in situ SEM imaging technique using the principle of thermal etching [5] was utilised to analyse the phase transformation of carbon steel during the heating stage of austenitic heat treatments. Both in situ SE and EBSD images were collected throughout the heat treatment process, which facilitated subsequent microstructural phase transformations characterisation from the in situ SEM data. The in situ evolution data were further supplemented by ex situ optical and EBSD data for both pre and post heat treatment. 

### 2.1. Materials

A 0.4 wt.% directionally drawn carbon steel, with a ferrite/pearlite phase microstructure, was selected for this study, which undergoes an austenitic phase change at 780 °C during heat treatment [6]. The study chose to focus on this phase transformation during the heating period as it significantly affects the final microstructural and mechanical properties that result from the heat treatment. Furthermore, the observation of key microstructural changes occurring during this evolution would facilitate improved control over this phase change to generate desirable properties from the heat treatment. 

### 2.2. In Situ SEM Heat Treatment

The phase transformations were observed during the in situ heat treatment of 0.4% carbon steel within the SEM. Samples were cut into disk shapes with an 8 mm diameter. To facilitate ease of phase change observation with respect to grain movement within an SEM, all samples were prepared by silicon carbide paper grinding followed by diamond suspension polishing and finished with a colloidal silica polish with a final thickness of 1 mm.

All in situ heat treatments of the specimens were facilitated by a purpose-built heat stage [5]. During the in situ heat treatment the initial microstructure was heated at a rate of 60 °C/minute up to 800 °C, which is within the austenitic region of 0.4% carbon steel [7]. The samples were subsequently held at this temperature for one hour before cooling at a rate of 20 °C/minute to 50 °C. The selected temperature provides a suitably slow phase transformation into austenite whilst still becoming fully austenitic. The speed of the transformation dictates the ease at which the phase evolution could be observed via SEM; a slower speed facilitated a greater number of images captured and thus a more detailed exploration of the microstructural phase transformation. 

SE images were captured, using a 15 kV electron beam, at regular intervals during the one-hour heat treatment, with a focus on the first 30–40 min, during which the main phase transformations, from ferrite/pearlite to austenite occurred. The different grayscale levels of the SE images allowed differentiation between different phases [8,9]. This phase information, coupled with thermal etching of the grain boundaries [5], was used to track the changes in the microstructural evolution with respect to phase and grain size. 

EBSD in situ data were also collected at 15-min intervals for one-hour, using Oxford Instruments Nano-analysis EBSD detector, during the holding stage of the heat treatment at 800 °C. The time interval between scans is due to a combination of the speed of map acquisition and the need to prevent the detector from overheating. To minimise detector overheating, the EBSD detector was retracted between scans to allow it to cool. Although the temperature of the detector did not exceed 55 °C during the heat treatment and the screen can be operated at temperatures up to 120 °C, it was observed that the processing of the data became significantly slower at elevated temperatures. The slowing of data processing was attributed to the location of the camera processing chip at the front of the EBSD detector, which was also subject to overheating. Scans were between 10 and 20 µm square with a step size of 0.35 µm and took between 2.5 and 4 min to collect; parameters were chosen as a compromise between the need to image enough grains for analysis and the necessity to perform the scan in a short time-period to capture the dynamic process. Post experimental analysis was conducted on the EBSD data to produce, grain, phase, and inverse pole figure maps.

### 2.3. Ex Situ Characterisation

To further supplement the in situ microstructural data captured at the phase transition temperature, optical images and EBSD scans were collected pre and post heating to confirm the significant change in microstructure attributed to the heat treatment process. Pre and post heating, samples were also polished and chemically etched using 2% Nital to facilitate optical imaging using an Alicona Profilometer. EBSD room temperature scans of the repolished surface were captured before and after heating at 100 µm square with a step size of 0.25 µm and scan time of 8 h using a 15 kV electron beam. 

### 2.4. Data Analysis of Phase Change Kinetics

To quantify the phase change rate, SE data at 500–1000× magnification with a distribution of >100 grains, which were visible owing to the presence of thermal etching, were processed using ImageJ analysis software. Histogram segmentation within ImageJ facilitated separation of the phases visible in the SE image, distinguishable by the different grayscale levels. Hence, the phase percentage present at each time interval, every 5–10 min over the course of one hour, was calculated. The results were subsequently plotted as the phase percentage against time to provide a rate of change of the phase transition from the moment that the temperature reached 800 °C. Understanding of the effects of phase atomic density combined with SE and EBSD images captured simultaneously enabled identification of which grayscale levels in the SE image were representative of each phase. 

The results of the phase change rate calculation were fitted to the phase transformation equation, the Johnson–Mehl–Avrami–Kohnogorov (JMAK) model [10],
f = 1 − exp(−Kt^n^)(1)
and were presented in its commonly rearranged form,
ln(ln(1/(1 − f))) = nlnt + lnK(2)
where f is the volume fraction of the transformed phase (in this case austenite), t is the transformation time in seconds, K is a temperature dependent constant defined by the Arrhenius type equation: k = B exp((−Q)/RT), Q is the activation energy, R is the gas constant, T is the temperature the phase transformation occurs at, B is a material constant, and n is the Avrami exponent of the equation [11]. The JMAK model is used to predict and describe the kinetics of isothermal phase transformations assuming transitions occur by nucleation and diffusion-controlled growth [11]. Thus, it is widely accepted as a good fit for the pure ferrite to austenite phase transition, in particular [10]. 

## 3. Results

### 3.1. Ex Situ Microstructures

Optical and EBSD images, presented in Figure 1, show the overall microstructure of the drawn steel before and after a one hour heat treatment at 800 °C. Focusing on the optically etched images, Figure 1a,b, the presence of ferrite (red arrow) and pearlite (blue arrow) phases are clearly visible. The phase change, as expected, has caused an increase in the size of larger grains and a decrease in the number of smaller ones, this can be slightly seen from the optical image. However, the use of EBSD to capture a localised area, Figure 1c,d, shows a significant change in grain size, indicating there has been grain growth during the phase change. The raw (uncleaned or raw EBSD images refer to those that have not undergone post processing after data capture) EBSD scans presented confirm that the heat treatment of ferrite/pearlite starting structure into the austenitic region led to the observation of grain growth ex situ. The significant microstructural change observed ex situ further demonstrate the need for in situ studies to understand the point during the heat treatment that this rearrangement occurs and how the phase transformation may affect this.

### 3.2. In Situ Phase Change: SE

In situ SEM imaging during the heat treatment enabled the documentation of the ferrite/pearlite to austenite phase change using both SE and EBSD imaging. Figure 2 shows a selection of SE images captured during the first 30 min of the heat treatment process. A significant variation in grayscale between images is observed, which is widely accepted to indicate the existence of different phases simultaneously in an SE image [12]; assuming no adjustment to the brightness or contrast was made between images. Therefore, the SE images presented in Figure 2 show the stages of the phase transition from a microstructural perspective. Figure 2a shows an SE image of the carbon steel surface once the required heat treatment temperature of 800 °C was reached. Here, an outline of the grain boundaries can be seen owing to the phenomenon of thermal etching. The outline also facilitates identification of individual grain’s phase presence, although this localised phenomenon is further investigated using EBSD high temperature imaging. Instead, the SE images can be used to examine the changing ratio of the three different grayscale levels corresponding to the phases, as the transformation of ferrite/pearlite to austenite occurs in real time.

Focusing on the evolution of the darker areas, these evolve dramatically in the first 10–15 min, with limited grain growth, but the dark areas appearing to turn to the mid grey shade, as identified in Figure 2b,c and highlighted by the red arrows. During this transition, some of the darker patches appear to fill gradually with the lighter grey spots, rather than the whole area transforming at once. The smaller dark grains change first ahead of the larger ones, this can be seen quite clearly in the difference between Figure 2b,d where almost all the smaller darker areas have transformed, shown by the red arrow in Figure 2d. It is important to note the impact of thermal etching on the image; in Figure 2d some of the darker areas appear to surround the grain boundaries, which can be accounted for by thermal etching presence showing the grain boundary outline. By 20 min, Figure 2f, all the darker areas have transformed to the mid grayscale phase and only the mid and light phases are present. 

Also observed in this sequence of SE images is the change of the lighter phase into the mid grayscale. However, this change is not observed until after 12 min, Figure 2d. This is then exacerbated, Figure 2e, after 16 min. Unlike the darker phase transformation, which occurs homogenously, the transformation to austenite appears to occur from the edges moving into the centre (heterogeneously)—as can be seen quite clearly in the larger, light grain shown by the green arrow in Figure 2d–f. The smaller lighter areas also begin to shrink quite rapidly and after 24 min (Figure 2g), these have almost all transformed. It is worth noting that Figure 2g also experiences some uncorrected beam drift and the green arrow highlights where the original large light grain is in the new position. It can be seen that this continues to shrink and by 32 min, there has been a complete transformation (Figure 2h). Noting the differences in the two-phase transformations, Figure 3 shows a focused transformation over the first 30 min of the specific dark phase cluster and Figure 4 presents a specific light phase cluster. 

### 3.3. In Situ Phase Change: EBSD

EBSD imaging was used to confirm the representation of the different phases within the SE grayscale images. Examination of the Kikuchi patterns of the different phases present at 800 °C, shown in Figure 5, identified that the mid grayscale areas were austenite and the lighter area was BCC ferrite. Hence, it may be assumed that the darker areas within the SE images represent the pearlite phase, the only other phase present during this transformation. This conclusion is also as expected based on the level of Carbon present in each of the phases and the representation of the structure during SEM imaging.

Grain evolution was also captured using EBSD scans, Figure 6, facilitating a localised understanding of the microstructural evolution during the phase transformation from ferrite/pearlite to austenite; all EBSD scans are taken in the same position unless stated otherwise. Figure 6a,b show there is a significant transformation in the orientation and size of the grains during the initial stages of heating from room temperature to 800 °C. There is also a change in phase, during which the small grains appear to transform prior to the larger ones. Examining the structure after 25 min, Figure 6c, shows a further change in orientation and complete phase change, indicating that the transformation process is complete; a similar timeline to the observations in the SE images presented in Figure 6. EBSD images of the same area at 40 min, Figure 6d, shows no change in phase and minimal growth of grains. The EBSD image taken after 57 min of heating, Figure 6e, shows some beam drift (see the black arrow for direction), but provides enough of the localised microstructure to suggest there is little further evolution.

Owing to the speed of the phase transformation, as seen in the EBSD images, the observation of the phase transformation using EBSD is limited to the first three scans in Figure 6. However, additional EBSD scans at this temperature, shown in Figure 7, further support the SE data. The scans indicate that as well as smaller grains transforming first, some initial nucleation of austenite grains occur within the ferrite/pearlite formation, and this is where transformation first occurs. Figure 7 depicts an initial room temperature scan (Figure 7a,b) and then a smaller scan in the same position after just 2 min of heating at 800 °C (Figure 7c,d). Where smaller grains have formed, the phase scan in Figure 7d indicates that these areas are austenitic (indicated by the blue) or alternatively concentrated areas of carbon (indicated by the red). The findings in Figure 7a, show an initial austenite nucleation between the pearlite grains; represented as a small agglomeration of ferrite grains on the right hand side of the image. Previous, rapid heat treatment followed by quenched cooling EBSD studies have indicated that at temperatures of 700–800 °C austenite nucleates preferentially as a block at triple junction at high angled grain boundaries [9]. The inverse pole figure data presented in Figure 5 also indicates this, where austenite is shown to occur in, and subsequently appears to undergo complete transformation of, these high angled grains first before transforming the lower angled grains. However, due to the similarities between pearlite and ferrite it is often difficult to disseminate the two phases using EBSD, hence the benefit of SE imaging to support these findings [13].

### 3.4. Tracking Phase Changes In Situ

The SE images were taken at sufficient frequency to facilitate the quantifiable tracking of the ferrite/pearlite to austenite phase change in situ. Percentages of the three phases quantified using ImageJ’s histogram segmentation tools were plotted against time. The combined results of three separate data sets are presented in Figure 8, on a log-linear graph, as is standard practice for phase transformations [4]. The graph demonstrates that the extreme grayscale ends (light and dark) representing ferrite and pearlite respectively decrease as the austenite phase increases and the graphs form a sigmoidal shape. 

To further compare these findings to other data-sets, the well-known JMAK model was used. The results plotted in Figure 9 show only the first 30 min of the data; once the phase change is complete, the model no longer applies. The graph presents two distinct linear regressions, highlighted by the black and red lines respectively, with a turning point between 10 and 12 min, circled. The two linear regressions, in the JMAK model, suggest two separate stages in the transformation process that are governed by two distinct systems of nucleation ad growth. The initial slow rate, and, hence, lower gradient correlation (black line in Figure 9), is attributed to the formation of nuclei of the new phase [12]. Once nuclei numbers reach a critical mass they begin to agglomerate leading to rapid growth of the new phase cluster [14]; demonstrated by the higher gradient graph (red line in Figure 9). This distinct difference in rate of the two step process means that when any nuclei begin to agglomerate, the growth phase takes priority even if some nuclei may still be forming [14], as can be seen in Figure 9. 

## 4. Discussion

Previous ex situ microstructural and dilatometry data indicate that the formation of austenite from a ferrite/pearlite microstructure is driven by nucleation and growth, where heterogeneous nucleation begins at the ferrite/pearlite grain boundaries before these nucleating sites begin to grow [2,15]. The most common dynamic model for this two stage process is the JMAK model [16] and by presenting the data in this familiar format; the data in Figure 9 supports this theory. 

Observations of the SE microstructural images indicate that the individual transformations of nucleation and growth are not solely associated with the two phases, but both phases experience the nucleation and growth process when transforming into austenite, as shown by Figure 2, Figure 3 and Figure 4. The SE images suggest that initial formation of austenite is predominantly focused in the pearlite region, as a combination of heterogenous and homogenous nucleation, followed by rapid growth of austenite nucleation within the pearlite grains, until complete transformation. Additionally, the ferrite to austenite transformation, which appears to begin between 10 and 12 min after the pearlite, could either be attributed to growth of the surrounding austenite grains into the ferrite or due to heterogeneous nucleation of austenite grains around the ferrite grain boundaries. These observations suggest the transformation of the ferrite and pearlite phases into austenite does not occur concurrently but has some overlap where both phases are transforming simultaneously. 

The complex nature of the starting structure resulting in the apparent overlap of pearlite and ferrite transition to austenite, like many dynamic processes, is dictated by the most energetically favourable transformation. In general, phase transformation into austenite is governed by diffusion of carbon into the newly formed austenite FCC phase, known as soft impingement [16]. The rate of carbon diffusion is dependent on the ease at which carbon can be extracted from its current phase and hence rearranged into the FCC austenitic phase. The pearlite phase is made up of a lamella of cementite (a carbon rich iron phase of Fe3C) and ferrite [3], whilst the pure ferrite phase is a simple BCC structure with interstitial carbon [17]. Thus, the initial nucleation within the pearlite grains, maybe due to the higher, less tightly bonded carbon content within the pearlite phase, results in a significantly higher carbon diffusion rate. Hence, nucleation formation of austenite within pearlite would occur prior to formation within ferrite grains. 

In further support of this observation, studies indicate that the rate of the reverse austenisation correlates positively to the number of boundaries serving as heterogeneous nucleation sites [18,19]. As such, the presence of lamella within the pearlite grains acting as boundaries would provide a more preferential, and subsequently faster, nucleation site than the uniform ferrite grains. This again indicates the possible reason for initial austenite formation within pearlite followed by ferrite. However, the phase transformations are not distinct, as observed in the current data, as the point at which the austenite nucleation clusters within the pearlite reaches such a peak as to lead to growth, may also be a similar time at which the system has sufficient energy to begin transformation of the ferrite grains. The nucleation of austenite at ferrite grain boundaries followed by growth of austenite into ferrite has been modelled and observed in previous studies of pure ferrite to austenite transformation [19,20]. As such, it is considered the likely process by which ferrite-austenite transformation occurs in the more complex pearlite/ferrite starting microstructure. 

## 5. Conclusions

This paper presents a study on the use of SE and EBSD in situ high temperature data to capture multi-grain carbon steel phase transformation. Consequently, the data provide both qualitative and quantitative insight into the mechanisms behind phase formation from ferrite/pearlite into austenite. The process observed within the SE and EBSD images confirm the pearlite/ferrite to austenite phase transformation models, which suggest a nucleation and growth driven process. Real-time microstructural evolution SE images indicate that the transformation of the two phases has significant overlap in time where nucleation of the pearlite phase begins initially in a homogeneous formation followed by heterogeneous nucleation around the pearlite/ferrite grain boundaries. Both phase nucleation processes lead to rapid growth of the nucleated austenite grains; first, in the pearlite, and then second, a growth from the edges to the centre in the ferrite grains until complete transformation. Quantification of the SE data in comparison to the JMAK model, combined with qualitative examination of both SE and EBSD images support this multi-phase nucleation and growth theory. Overall, the findings presented in this paper provide an alternative technique for capturing areas undergoing microstructural evolution during a phase change, which are representative of the bulk of the specimen. The data captured using these techniques enable clarification on the timing and position of austenite formation within a ferrite/pearlite starting structure and, hence, understanding of how to adjust heat treatments to produce more favourable microstructural properties.

## Figures and Tables

**Figure 1 materials-15-03921-f001:**
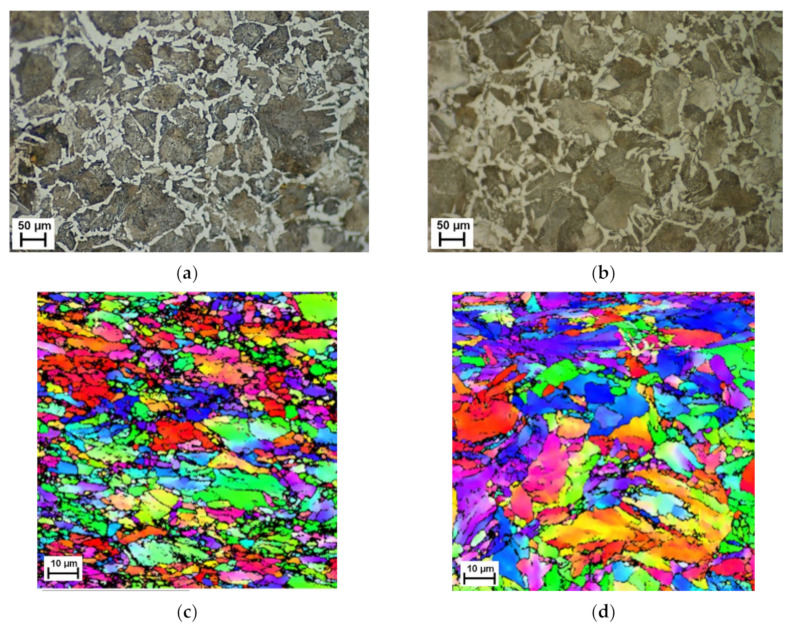
Optical (**a**,**b**) and EBSD (**c**,**d**) images at room temperature before (**a**,**c**) and after (**b**,**d**) heat treatment of ferrite/pearlite starting structure specimens.

**Figure 2 materials-15-03921-f002:**
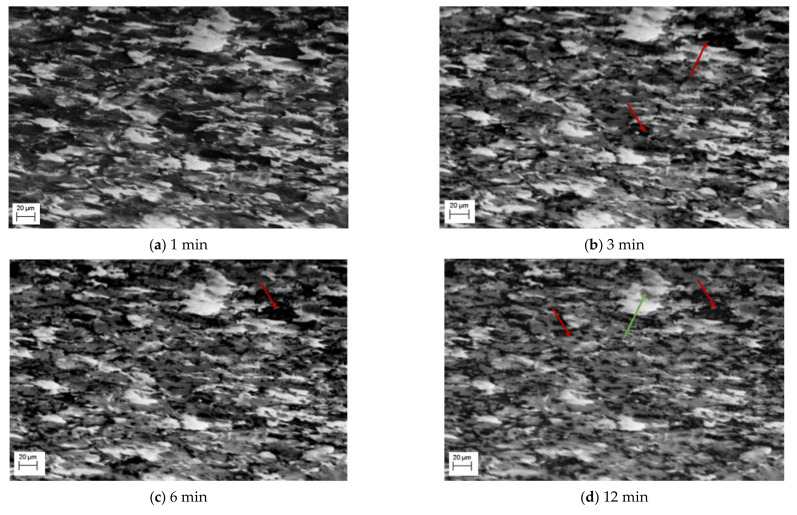
SE images during ferrite/pearlite to austenite phase change after heating at a temperature 800 °C indicates a multi-phase transformation of nucleation and growth. The red arrows highlight the transformation of a single dark grain which undergoes transformation and the green arrows a single light grain.

**Figure 3 materials-15-03921-f003:**
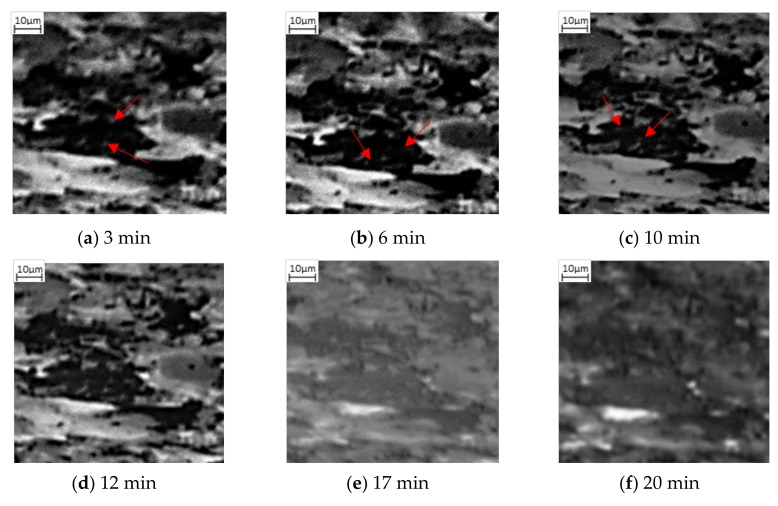
Phase transformation of a darker grain into the mid grayscale form via nucleation in (**a**–**c**) followed by growth (**d**,**e**) and complete transformation (**f**). The red arrows show initial lighter grey spots forming within the dark phase, increasing in numbers from 3 min after heating (**a**) through to 10 min of heating (**c**).

**Figure 4 materials-15-03921-f004:**
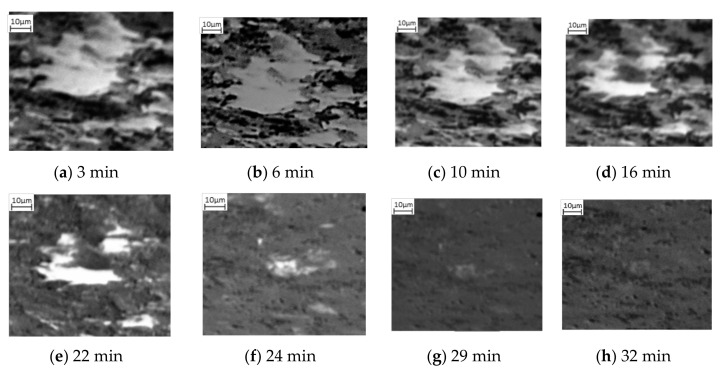
Phase transformation of a lighter grain into complete mid grayscale form where no visible transformation occurs between (**a**,**b**). Homogeneous nucleation is observed at 12 min after heating (**c**) and continues to grow through the first 16 min (**d**), 22 min (**e**), 24 min (**f**), 29 min (**g**) after heating until complete agglomeration at 32 min (**h**).

**Figure 5 materials-15-03921-f005:**
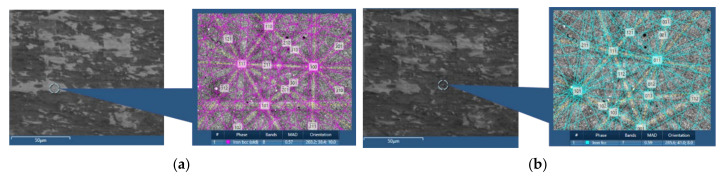
EBSD Kikuchi patterns, taken at 800 °C, identifying the phases present from the SE image. (**a**) Lighter area identified as BCC ferrite and (**b**) darker area identified as FCC austenite.

**Figure 6 materials-15-03921-f006:**
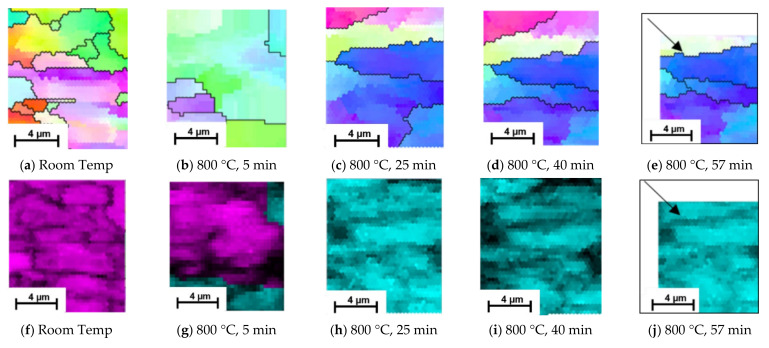
Evolution of ferrite/pearlite to austenite captured at room temperature before heating (**a**,**f**) and after heating at 800 ℃ for 5 min (**b**,**g**), 25 min (**c**,**h**), 40 min (**d**,**i**) and 57 min (**e**,**j**) via EBSD Inverse Pole Figure (**a**–**e**) and Phase (**f**–**j**) maps.

**Figure 7 materials-15-03921-f007:**
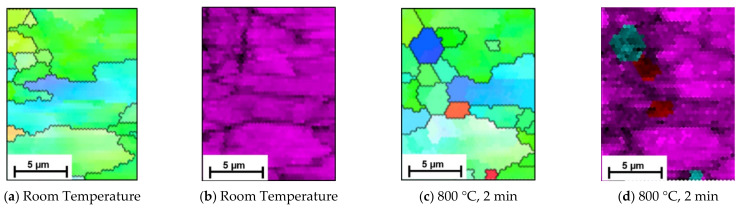
EBSD scans represented by the Inverse Pole Figure (IPF) (**a**,**c**) and corresponding phase maps (**b**,**d**) capture the preliminary formation of austenite within a ferrite/pearlite microstructure when heated to 800 °C within an SEM. (**a**,**b**) show the initial area, and (**c**,**d**) show the same area where the phase transformation occurs and austenite (in blue) has begun to agglomerate and form in the smaller grains.

**Figure 8 materials-15-03921-f008:**
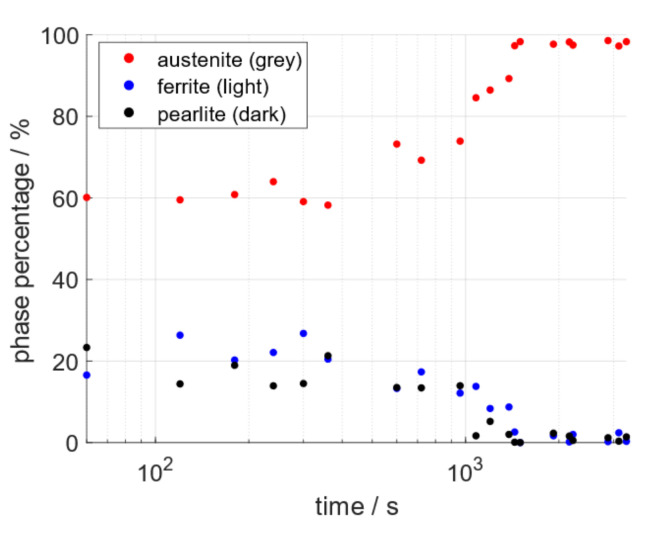
Offset sigmoidal plot, the data for which were quantified from the SE images of three data sets showing the phases: austenite, ferrite, and pearlite as red, blue, and black respectively on log-linear plot.

**Figure 9 materials-15-03921-f009:**
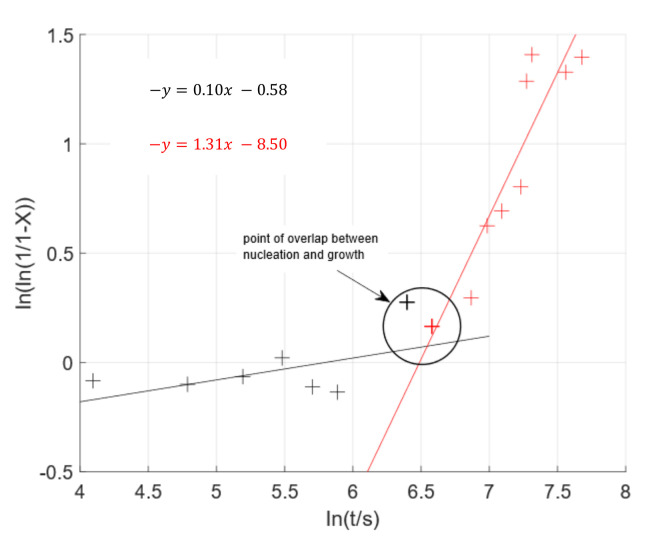
JMAK model applied to the SE data collection during the ferrite/pearlite to austenite phase transformation at 800 °C. The model indicates two distinct kinetics of nucleation and growth governing the phase transition.

## Data Availability

All data is provided within the article.

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
