# Peer review of "Investigating Iron Alloy Phase Changes Using High Temperature In Situ SEM Techniques"

_materials, 2022, doi:10.3390/ma15113921_

Round 1
Reviewer 1 Report
The authors present a manuscript monitoring by in situ SEM technique, through SE and EBSD studies, the phase transformation from ferrite/pearlite into austenite (fcc) observed with a one hour heat treatment process at 800ºC. Process follows the JMAK model, namely a slow initial homogeneous nucleation of pearlite followed by a heterogeneous nucleation around the pearlite/ferrite grain boundaries. These lead to a rapid growth of the nucleated austenite grains, firstly in the pearlite and secondly form the edges of the ferrite grains. This study is interesting due to its in situ character; being also demonstrated by previous works of the authors.
Moreover, manuscript is well-written and easy-to-read. However, despite the scientific content is good, correct and well-explained, there are some points related to article’s format that should be addressed by authors. Thus, IRECOMMENDED IT FOR PUBLICATION AFTER MINOR REVISION.
The points that should be addressed are the following:
1.- Materials and Methods section, In Situ Heat SEM Heat Treatment subsection, line 83; authors write “purpose-built heat stage” referring it to Reference [7]. This work is theoretical. I think the correct is the Reference [5]. This is a previous work of the authors, in the same thematic of the work and where a heat stage image is present. Please, confirm and correct text.
2.- References; please introduce within the text between brackets, not with parenthesis.
3.- Figures, scale bars; Scale bars of Fig. 1 a &b, Fig. 3 are little and difficult to see. On the other hand, scale bars for Figs. 6 and 7 are high, but the contrast is not good. Considering all figures of the manuscript, I recommend standardizing scale bars to model of Figs. 2, 3 and 4. This is the right one.
4.- Figures, Fig. 2, arrows; I suggest enlarging size and, furthermore, enhance contrast of arrows to facilitate their observation and, consequently, comprehension of the readers.
5.- Figure captions and Manuscript body; please write ‘&’ symbol along the text instead of that noted, f.ex., on Fig. 1 caption. Maybe it is a mistake induced by text corrector program.
6.- Figure 7; Please change color of ferrite or pearlite points to distinguish well between them. Maybe, green. Likewise, increase size of the graphic to enhance comprehension.
7.- Figure 8; I recommend standardizing scale points and legends to those of the Fig. 7 with the aim to facilitate comprehension and enhance quality. Thus, change text size and, also, put letters/numbers in bold. I also suggest increasing the size of the figure.
Author Response
Thank you very much for your very complimentary review. I have addressed the points raised as follows:1) Thank you for noticing this, I can confirm this is correct and I have since checked all references to ensure the correct citation is provided.
2) Referencing style has been updated accordingly.
3) The scale bar has now been standardised across all microscopy images.
4) Colour of arrows adjusted but size remains the same so as not to overshadow the image. Possibility of making image larger in the paper if space in the journal allows. The current reason for image size is owing to page limit.
5) This is due to the standardised MDPI font and cannot be changed.
6) The colours cannot be easily changed but have adjusted contrast so that it is more visible.
7) The size is limited due to the page count specified by MDPI.
I would like to once again thank the reviewer for taking the time to read and review the paper.
Reviewer 2 Report
Congratulations, it is a great work and an excellent manuscript. The heat stage combined with a Zeiss scanning electron is a great technical achievement. The vacuum within the SEM sample chamber prevent the oxidation during heating allowing proper observation of phase transformation. Investigation of austenitic transformation in carbon steel was a smart choice to illustrate the experimental method ability. EBSD and SE methods give a complex characterization of the sample microstructure. The used working regime allows optimal investigation of pearlite and ferrite transformation into austenite kinetics. Image J soft is adequate for the microstructure quantification.
The experimental setup is well designed and the obtained results are relevant and support the discussion and the conclusions.
The manuscript is well documented with relevant references: some of them are older but very important as fundamental background and some of them are newest figuring out the actual trend in the research field.
A recommendation is to make the images scale bar sharper and the font a little bit more visible.
Author Response
Thank you very much for taking the time to read and review the paper.
Your positive comments were very well received.
I have now standardised the scale bars for Figures 2,3,4 so this should be easier to read.